# Adaptation of Generalist Robot Policies with Minimal Data

Shreyas Kowshik*, Sreyas Venkataraman*, Leo Wang, Niharika Pant, Max Simchowitz†, Aviral Kumar†
**Carnegie Mellon University**

*Abstract*—**A central goal of robot learning is to move beyond task-specific human data collection toward robots that improve through autonomous interaction. Yet fully autonomous learning remains difficult with current policies: sparse rewards and weak zero-shot exploration make it unlikely that a robot will discover successful behavior from scratch. We study minimal-data adaptation, a regime in which a pretrained robot policy must learn a new task from as little as one demonstration followed by autonomous online interaction. This setting minimizes task-specific supervision while testing whether pretrained policies provide enough structure to make reinforcement learning tractable. We present MIDAS , an offline-to-online RL framework that first anchors a pretrained VLA to the target task with few-demo behavior cloning, then improves it through value-based online residual RL. Across LIBERO and RoboCasa, MIDAS recovers strong task performance from as little as one demonstration, substantially outperforming baselines and generalizing beyond demonstrated conditions. To the best of our knowledge, this is the first demonstration of reliable robot policy adaptation from a single task demonstration. We further validate MIDAS on bimanual YAM hardware, where it learns effective behaviors with roughly six hours of autonomous interaction.**

## I. INTRODUCTION

A long-standing goal in robotics is to deploy robots that improve through their own experience: a robot enters a new environment, attempts a task, and refines its behavior until it succeeds reliably, without further human supervision. While autonomous adaptation faces challenges at several levels, in this work, we focus on the algorithmic barriers to fast robot adaptation to a new task. Prior work has studied autonomous adaptation largely with specialist, single-task policies [20, 12, 28], often requiring complex task-specific methodology. The prevalence of generalist robot policies [27, 6, 5, 15, 4, 31, 16], pretrained on Internet data makes it natural to revisit this question: can a broad pretrained prior make online adaptation possible from minimal task-specific supervision?

To answer this question, we first note that autonomous adaptation requires three ingredients. The policy must represent high-dimensional observations in a form useful for control and value learning, produce behavior that reaches reward-bearing regions during deployment, and use this experience to improve efficiently. Recent generalist robot policies, including vision-language-action (VLA) models, make progress on the first two ingredients [27, 6, 5, 15, 4, 16]: their representations provide useful features for control, and their action heads encode behavioral priors learned from diverse manipulation

data. However, these policies are still not reliable zero-shot learners in new environments. They may fail to follow the task instruction, miss sparse reward entirely, or visit only a narrow set of useful states, leaving online learning with too little signal to bootstrap improvement.

We study this gap by proposing the ***minimal-data adaptation (MDA)*** setting, in which a pretrained policy must learn to improve autonomously from only minimal task-specific supervision, such as a single successful demonstration. Prior work has developed many of the tools needed for this setting, but typically assumes either that the initial policy already has substantial task competence [29, 25, 14] or that tens to hundreds of demonstrations are available [18, 22, 14, 3]. Few-shot imitation learning methods suggest that learning from one or two demonstrations is possible in some settings [8, 9, 24], but most available VLA policies do not naturally exhibit reliable few-shot improvement (related work discussed in Appendix B). MDA targets the narrow regime between zero-shot deployment and standard task-specific finetuning: the pretrained policy is not reliable enough to deploy directly, but may be close enough that one demonstration can make online learning tractable.

Our **central finding** is that minimal-data adaptation is already tractable for modern generalist robot policies. Even just a single successful demonstration can make autonomous adaptation possible, not because one demonstration provides enough coverage for imitation learning, but because adaptation succeeds through a division of labor across different pieces. Pretraining provides reusable features and broad action priors, but these priors are not calibrated for reliable performance on the target task. The demonstration anchors the policy to the task and makes task-relevant behavior reachable, but does not provide reliable control. Autonomous interaction supplies the remaining corrections by learning from the robot's successes and failures. We instantiate this principle in **MIDAS** : an offline-to-online recipe that first finetunes a pretrained VLA by imitation on $K$ demonstrations to obtain a task-anchored policy and then freezes the VLA but trains a lightweight residual actor online on top of its representations using value-based RL [21] on data collected via autonomous interaction.

We make three contributions in this paper. **First**, we introduce *minimal-data adaptation*, a regime where a pretrained robot policy is not reliable zero-shot, but may be close enough that one or a few demonstrations make online learning tractable. **Second**, we present MIDAS, a value-based deep RL method that combines few-demo behavior cloning with frozen-

*Equal contribution. †Equal advising.
Corresponding Authors: {skowshik, sreyasv, msimchow, aviralku}@andrew.cmu.edu;

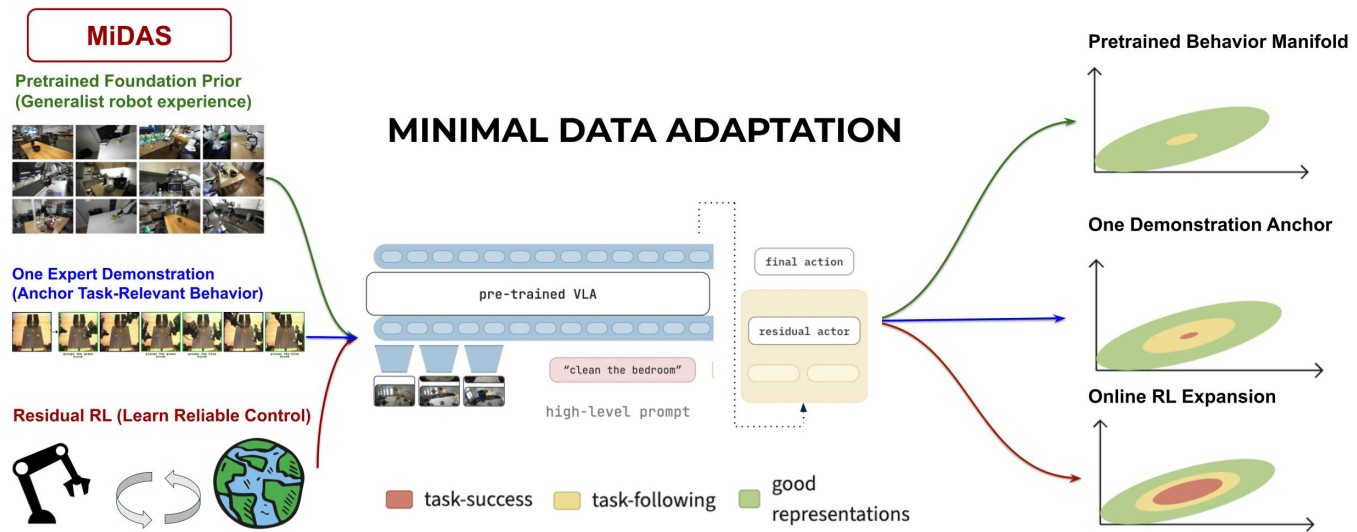

**Figure 1: Overview of minimal-data adaptation .** A single demonstration is sparse and underdetermined on its own, but pretrained VLA representations turn it into a useful anchor for adaptation. Few-demo behavior cloning narrows the search space to a task-relevant region around the demonstration, and online residual RL then expands local success by learning value-guided corrections on top of the warm-started base policy.

backbone residual RL and stabilizing choices for sparse-reward adaptation. Across LIBERO-Long and RoboCasa, MiDAS recovers strong performance from as little as one demonstration, outperforms baselines, generalizes beyond demonstrated conditions, and transfers to a real bimanual YAM platform with roughly 6 hours of autonomous interaction. **Third**, we use this regime to explain why adaptation works: behavior cloning recovers task-level behavior but not reliable control, pretrained VLA representations enable sample-efficient value learning, and online residual RL supplies corrective actions beyond the few-demo policy's effective support. Together, these results show that modern pretrained robot policies can use a single demonstration to scaffold autonomous adaptation, while clarifying what how pretraining, task specification, and online experience determine adaptation ability.

## II. PROBLEM SETTING: MINIMAL DATA ADAPTATION

We now define the minimal-data adaptation (MDA) setting and motivate why it is a useful regime to study. Given a pretrained vision-language-action policy $\pi_{\text{base}}$, $K$ successful demonstrations $\mathcal{D}_{\text{demo}}^K$ collected in a target environment (denoted by $\mathcal{M}_{\text{target}}$), and access to autonomous interaction with $\mathcal{M}_{\text{target}}$, minimal-data adaptation asks whether we can learn an adapted policy $\pi_\theta^K$ that achieves high return in $\mathcal{M}_{\text{target}}$ using only this limited supervision. The goal is not to solve the task from demonstrations alone. Instead, the demonstrations serve as a minimal scaffold that makes subsequent autonomous learning tractable. The case $K = 1$ is the sharpest version of this setting: learning on a single demonstration is often insufficient to learn previously unseen tasks, but may still expose enough task-relevant behavior for interaction-driven improvement to begin.

MDA targets the narrow regime where autonomous learning first becomes tractable. With no task-specific data, current pretrained policies often fail to reach task-relevant behavior

often enough for sparse-reward interaction to improve them. A small number of demonstrations can make the problem just structured enough: they expose the intended behavior and bring the policy into the vicinity of success, while still leaving robustness and distribution coverage to be learned through autonomous interaction. This setting therefore tests whether pretrained robot policies have reached a regime where a few demonstrations can specify the task while online interaction supplies the missing robustness. It also separates the roles of the available sources of supervision: the structure inherited from pretraining, the task information provided by demonstrations, and the corrections learned through autonomous interaction. Our approach MiDAS uses value-based residual RL to perform this online refinement; we briefly review these primitives before describing the full adaptation recipe.

## III. PRELIMINARIES

We formalize MDA in the standard reinforcement learning setting. The goal is to learn a policy $\pi : \mathcal{S} \to \Delta(\mathcal{A})$ that maximizes the discounted return $J(\pi; \mathcal{M}) = \mathbb{E}^{\mathcal{M},\pi}[\sum_{t=0}^\infty \gamma^t r(s_t, a_t)]$, where $s_0 \sim \rho$, $a_t \sim \pi(\cdot \mid s_t)$, and $s_{t+1} \sim P(\cdot \mid s_t, a_t)$. We focus on sparse binary rewards, where $r_t = 0$ if the task is completed within the time limit and $-1$ otherwise. A central object in our method is the action-value function $Q^\pi(\mathbf{s}, \mathbf{a})$, which estimates the expected discounted return from taking action $\mathbf{a}$ in state $\mathbf{s}$ and following $\pi$ thereafter. In our robotic setting, states include high-dimensional visual and proprioceptive observations, and actions are temporally extended chunks [32]: each $\mathbf{a}_t$ denotes a sequence of low-level controls executed open-loop. See Section A for details.

**Value-Based RL.** An approach to value-based RL maintains a replay buffer $\mathcal{B}$ of previously collected transitions and trains a critic using temporal-difference (TD) learning. Given a

transition $(\mathbf{s}_t, \mathbf{a}_t, r_t, \mathbf{s}_{t+1}) \sim \mathcal{B}$, the TD target and loss are

$$y_t = r_t + \gamma Q_{\bar{\psi}}(\mathbf{s}_{t+1}, \mathbf{a}'),$$
$$\mathcal{L}_{\mathrm{TD}}(\psi) = \mathbb{E}_{\mathcal{B}}\left[(Q_\psi(\mathbf{s}_t, \mathbf{a}_t) - y_t)^2\right], \qquad \text{(III.1)}$$

where $Q_{\bar{\psi}}$ is a target critic and $\mathbf{a}'$ is the next action. The policy $\pi_\theta$ is then updated toward actions that receive high value under the critic. [10, 11]

**Residual RL.** Residual RL adapts a frozen base policy by learning a lightweight policy module that modifies the base policy's proposed action [7, 30, 26, 1, 2]. Given a base policy $\pi_\theta$, the residual policy $\pi_\theta^{\mathrm{res}}$ conditions on both the state and an action proposed by the base policy, and outputs the action executed in the environment. This residual update can be parameterized in several ways, such as predicting an additive offset in action space or directly predicting the modified action. In this work, we adopt the latter parameterization as discussed in Section IV.

**Policy-Agnostic RL.** Policy-agnostic RL (PA-RL) [21] is an online improvement method that applies across policy classes. It first improves sampled actions against the learned Q-function, then distills the optimized actions back into the parametric policy using the appropriate supervised objective, such as flow matching for flow policies or likelihood for discrete-action policies. We use PA-RL for this policy-agnostic structure, extending it to action-chunked residual policies in Appendix E-A. For each replay state $\mathbf{s}_t$, we sample candidate action chunks, refine them by optimizing directly against $Q_\psi$ to obtain $\mathbf{a}_t^\star$, and distill the optimized chunk back into the residual policy.

## IV. MiDAS : A Simple Recipe for Minimal Data Adaptation

In this section, we introduce **Mi**nimal-**D**ata **A**daptation **S**trategy (MiDAS ), a two-stage recipe for adapting a pre-trained policy to a new task from only a handful of demonstrations. What can such minimal supervision realistically provide? As shown in Section V and depicted in Figure 1, finetuning a pre-trained model with $K$ initial demonstrations seeds coarse task-following behavior, but does not ensure reliable completion. This perspective leads to a simple two-stage recipe. **Stage I** fine-tunes $\pi_{\mathrm{base}}$ on the $K$ demonstrations with behavior cloning, producing a policy $\pi_{\mathrm{base}}^K$ that can attempt the correct task but remains unreliable. **Stage II** freezes $\pi_{\mathrm{base}}^K$ and trains a lightweight residual actor-critic on top of its representations using sparse-reward online RL. The resulting policy, $\pi_{\mathrm{RL}}^K$, provides the fine-grained control corrections needed for a high-success rate policy.

**Stage I: Behavior cloning on minimal demonstrations.** Stage I adapts the pretrained policy $\pi_{\mathrm{base}}$ to the downstream task using the $K$ demonstrations in $\mathcal{D}_{\mathrm{demo}}^K$. We adapt the VLM backbone using a low-rank adapter (LoRA [13]) for compute efficiency, and fine-tune the complete action head using the standard flow matching objective. Implementation details are deferred to Section E-B. As described in Section V,

the resulting $\pi_{\mathrm{base}}^K$ coarsely attempts the target task, but does not complete it reliably.

**Stage II: Online improvement via value-based residual RL.** Stage II improves $\pi_{\mathrm{base}}^K$ through autonomous interaction using value-based RL. Because Stage I produces a policy that can roughly complete the tasks, Stage II need only provide small corrections to overcome control failures. Therefore, to anchor RL updates around $\pi_{\mathrm{base}}^K$, we freeze the pretrained policy backbone and train only a lightweight adaptation module on top of it. Following common practice in prior work [7, 30, 26, 1, 2], we instantiate this module as a policy that predicts a correction to the action proposed by $\pi_{\mathrm{base}}^K$. At each state $\mathbf{s}_t$, the base policy proposes an action chunk $\mathbf{a}_t^{\mathrm{base}}$, and the residual policy $\pi_\theta^{\mathrm{res}}$ proposes an edit conditioned on the state and the chunk itself. We parameterize the residual policy as follows[1]

$$\pi_\theta^{\mathrm{res}}(\cdot \mid \mathbf{s}_t, \mathbf{a}_t^{\mathrm{base}}) = \tanh\left(\mathcal{N}\left(\mu_\theta(\mathbf{s}_t, \mathbf{a}_t^{\mathrm{base}}),\ \sigma_\theta^2(\mathbf{s}_t, \mathbf{a}_t^{\mathrm{base}})\right)\right). \quad \text{(IV.1)}$$

Conditioning the actor on $\mathbf{a}_t^{\mathrm{base}}$ exposes it directly to the action mode selected by $\pi_{\mathrm{base}}^K$, so online RL refines a task-relevant VLA proposal rather than learning an action distribution from scratch. At the same time, predicting $\mathbf{a}_t^{\mathrm{exec}}$ directly allows the policy to shift this proposal toward higher-value regions instead of being restricted to small fixed residual corrections around $\mathbf{a}_t^{\mathrm{base}}$ that further limit the policy. Then, we evaluate this action by a critic ensemble $\{Q_{\psi_j}(\mathbf{s}_t, \mathbf{a}_t^{\mathrm{exec}})\}_{j=1}^N$ trained with temporal difference learning in Eq. (III.1). We discuss implementation details for this residual parameterization in Appendix Section C.

**Overcoming the challenges introduced by minimal demonstrations.** Given minimal demonstrations, few expert trajectories exist for initializing the learned critic and residual-actor. Moreover, autonomous rollouts from $\pi_{\mathrm{base}}^K$ succeed only rarely, for example 2-5% of the time. Lastly, minimal demonstrations need not ensure that $\pi_{\mathrm{base}}^K$ contains optimal action in its support (Section V). We now introduce key design choices to overcome these respective challenges.

**1) Offline warmup: residual actor initialization and critic calibration from few demonstrations.** A randomly initialized residual actor would produce actions unrelated to the base policy, making early exploration unstable. We therefore construct a "warmup" buffer $\mathcal{B}_{\mathrm{warm}}$ from the $K$ demonstrations in $\mathcal{D}_{\mathrm{demo}}^K$ together with a small number of autonomous rollouts from $\pi_{\mathrm{base}}^K$ that may or may not be successful. For each state in this buffer, we query the frozen base policy to obtain $\mathbf{a}^{\mathrm{base}} \sim \pi_{\mathrm{base}}^K(\cdot \mid \mathbf{s})$, and train the residual actor to reproduce this base proposal:

$$\mathcal{L}_{\mathrm{warm}}(\theta) = \mathbb{E}_{(\mathbf{s}, \mathbf{a}^{\mathrm{base}}) \sim \mathcal{B}_{\mathrm{warm}}}\left[\left\|\mu_\theta(\mathbf{s}, \mathbf{a}^{\mathrm{base}}) - \mathbf{a}^{\mathrm{base}}\right\|_2^2\right]. \quad \text{(IV.2)}$$

This initialization makes the residual actor approximately preserve the behavior of $\pi_{\mathrm{base}}^K$: Online RL also requires a critic that can guide improvement from the start. While prior work can pretrain such critics from offline data [18, 22, 34, 17], our

---

[1]Unlike typical residual formulations, (IV.1) directly predicts the edited action $\mathbf{a}_t^{\mathrm{exec}} \sim \pi_\theta^{\mathrm{res}}(\cdot | \mathbf{s}_t, \mathbf{a}_t^{\mathrm{base}})$.

**Table I: One-demo adaptation results measured over 3 seeds.** Success rates (%) are mean ± std. Robocasa mean ignores last task.

| Task | 1 Demo | | | |
|---|---|---|---|---|
| | BC | DSRL | Filt. BC | **MiDAS** |
| *LIBERO-Long* | | | | |
| Alph. Soup + Cr. Cheese → Basket | 12.7±1.2 | 12.0±2.0 | 14.0±2.0 | **99.3**±1.2 |
| Alph. Soup + Tom. Sauce → Basket | 4.0±3.5 | 3.3±2.3 | 2.0±0.0 | **82.7**±13.3 |
| Black Bowl → Bottom Drawer | 22.7±6.4 | 28.7±4.2 | 28.7±10.3 | **98.7**±1.2 |
| Book → Caddy | 37.3±7.0 | 42.7±6.1 | 28.0±2.0 | **98.7**±2.3 |
| Both Moka Pots → Stove | 13.3±3.1 | 14.7±4.6 | 19.3±7.0 | **76.0**±14.0 |
| Cr. Cheese + Butter → Basket | 34.0±9.2 | 28.0±2.0 | 68.0±2.0 | **96.0**±0.0 |
| Moka Pot → Stove | 84.0±0.0 | 92.0±0.0 | 85.3±8.3 | **95.3**±3.1 |
| White Mug + Choc. Pudding | 33.3±11.7 | 45.3±6.4 | 42.0±4.0 | **96.7**±5.8 |
| White Mug + Plates | 30.0±2.0 | 17.3±1.2 | 34.7±6.4 | **79.3**±11.0 |
| Yellow-White Mug → Microwave | 64.0±2.0 | 50.7±4.6 | 68.0±7.2 | **89.3**±8.1 |
| **Average** | 33.5±1.8 | 33.5±0.8 | 39.0±5.9 | **91.2**±4.2 |
| *RoboCasa* | | | | |
| Banana: Fridge Drawer → Shelf | 13.3±3.1 | 12.7±2.3 | 19.3±3.1 | **93.3**±7.0 |
| Hot Dog: Counter → Cabinet | 29.3±1.2 | 20.0±10.4 | 36.0±24.0 | **100.0**±0.0 |
| Mug → Coffee Machine + Start | 24.0±8.7 | 24.7±4.2 | 46.7±6.2 | **74.7**±20.0 |
| Cup + Bowl → Dishwasher + Close | 0.0±0.0 | 0.0±0.0 | 0.0±0.0 | 0.0±0.0 |
| **Average** | 22.2±3.4 | 19.1±3.2 | 34.0±10.6 | **89.3**±8.7 |

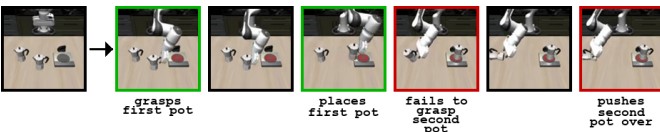

**Figure 2: Few-demo BC recovers task-level behavior, but not fine-grained control.** Rollouts of $\pi_{\text{base}}^K$ at $K=1$ on *Both Moka Pots → Stove*. The policy reaches the right objects but fails to grasp the second pot under orientation changes.

sources of supervision that enable MIDAS to achieve this: First, the few-demo BC finetuning of the pre-trained VLA provides actions which follow task instructions coarsely, reducing the burden of exploration. Second, the pretrained VLA provides visual representations, enabling efficient learning without direct state access. Finally, online reinforcement fills in the gaps in precise-control, which necessarily requires value-guided online updates to finetuning actions beyond what $\pi_{\text{base}}^K$ can reliably produce on its own. Additional ablations of MIDAS are deferred to Appendix F and experiments to Appendix I.

**Experimental setup.** We use $\pi_{0.5}$ [15] as the pretrained base policy $\pi_{\text{base}}$ and run MIDAS with $K=1$ demonstration only on two simulation benchmarks: **a)** LIBERO-Long [19], which studies long-horizon language-conditioned manipulation, and **b)** RoboCasa-365 [23], which studies long-horizon household manipulation in diverse kitchen scenes (Details in Appendix D-A). By ablating different components of our approach, we aim to identify the core mechanisms that make MIDAS work well.

**1) Pretraining and task-specific demonstrations enable coarse behavior recovery.**

Before online adaptation, we ask whether $K=1$ can recover task-anchored behavior. Figure 3 compares $\pi_{\text{base}}^1$ to three approaches: **a)** a flow-matching policy trained on pretrained ResNet features, which tests whether generic

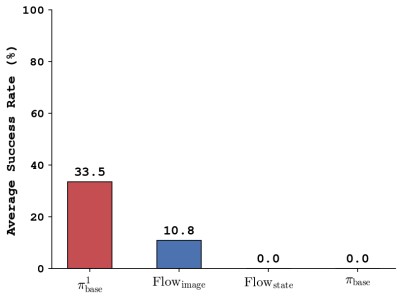

**Figure 3: Performance of different task representations for behavior cloning**

setting has no additional offline data beyond the few demonstrations and autonomous rollouts from $\pi_{\text{base}}^K$. We therefore draw inspiration from Zhou et al. [34] and train the critic on the warmup buffer $\mathcal{B}_{\text{warm}}$ using TD learning (Eq. (III.1)). Importantly, this critic is trained on the state-action distribution induced by $\pi_{\text{base}}^K$, making it calibrated to the regions where online improvement begins and limiting forgetting.

**2) Success Balancing: overcoming limited initial policy performance.** Since the base policy may only succeed 2-5% in some cases, we employ *success balancing* to amplify the sparse reward signal during both the warmup and online improvement phases. To do so, we maintain a standard replay buffer $\mathcal{B}$ containing all collected trajectories and a success buffer $\mathcal{B}_{\text{succ}}$ containing only successful trajectories, both initialized with the same $K$ demonstrations. Critic minibatches are sampled from a mixture of these buffers, oversampling $\mathcal{B}_{\text{succ}}$ to prevent the value function from being dominated by failed rollouts and to guide policy improvement toward reward-bearing regions. Further details are present in Appendix E-C.

**3) Residual policy training with PA-RL [21].** Lastly, Section V reveals the need for actions that go beyond the support of the base policy. While many value-based RL algorithms can provide this, we opt for "policy-agnostic" PA-RL approach, which we apply to the residual actor. To do so, we sample *a single* proposal action $\mathbf{a}_t^{\text{base}}$ from the base policy, then apply PA-RL style Best-of-N distillation objective with an action gradient to a batch of $N$ actions $\mathbf{a}_t^{(i)} \sim \pi_\theta^{\text{res}}(\cdot \mid \mathbf{s}_t, \mathbf{a}_t^{\text{base}})$ sampled from the residual. See Appendix E-A for more details. Importantly, sampling requires *only* a single expensive forward pass through $\pi_{\text{base}}^K$ and no backward passes since the base is frozen.

## V. WHAT ENABLES MIDAS TO LEARN EFFECTIVELY?

Previously, we motivated MIDAS with the premise that limited demonstrations can seed behavior close enough to success to to seed online improvement. In this section, we present detailed empirical evidence identifying how the three

visual pretraining is sufficient; **b)** a flow-matching policy trained on privileged state inputs, which tests whether near-optimal state information alone enables recovery of task-anchored behavior; and **c)** the zero-shot pretrained policy $\pi_{\text{base}}$, which tests whether pretraining alone identifies the task. All three perform poorly: the ResNet flow-policy achieves non-zero success on only 3/10 LIBERO-Long tasks, while the privileged-state policy and $\pi_{\text{base}}$ obtain 0% success. Together, these failures suggest that MDA requires both task grounding from the demonstration and behavior-aware representations from pretraining; privileged state inputs alone do not recover this behavior, suggesting that the benefit of pretraining goes beyond improved state representation.

Qualitative rollouts in Fig. 2 show what $\pi^1_{\text{base}}$ recovers despite its low success. The policy often approaches the correct object, attempts the correct subtask sequence, and follows a trajectory resembling the demonstration. Its failures are primarily in reliable execution: contact, alignment, grasping, and recovery remain brittle. Thus, pretraining and one demonstration learn coarse task behavior, but online adaptation is needed to solve the control problem.

**2) Pretrained representations simplify online adaptation and substantially improve sample-efficiency.** The preceding results show that large-scale pretraining, together with minimal demonstrations, anchors the search for task-relevant behavior. We now find that pretrained VLA representations also simplify the subsequent online learning problem, making sparse-reward residual RL substantially more sample-efficient. Figure 4 isolates this effect by varying the representation used by the residual learner. With a ResNet fine-tuned from scratch, MIDAS fails to converge at $K=1$, indicating that sparse interaction is insufficient to learn both task-relevant visual features and fine-grained action priors for control. Generic visual pretraining helps, but frozen DINO features still require substantially more interaction, suggesting that visual invariances alone are not enough to serve as good pretrained features. In contrast, features from the pretrained $\pi_{0.5}$ backbone enable rapid improvement, suggesting that VLA pretraining yields representations already aligned with robot control.

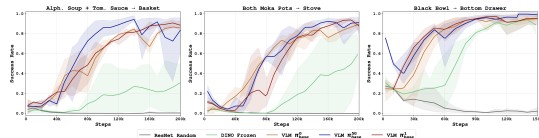

**Figure 4: Performance of different representations in online adaptation.**

**3) Value-based RL learns fine-grained control actions beyond the base policy's prior.** The previous analysis showed that $\pi^K_{\text{base}}$ often produces task-relevant behavior, but solves the task inconsistently. We ask: do the right actions already lie within its support but are not reliably selected, or do they fall outside its effective support?

We test this by comparing MIDAS to two representative alternatives: *Filtered BC*, which finetunes $\pi^K_{\text{base}}$ on successful online rollouts and sharpens modes the policy already reaches; and *Diffusion-Steering RL* (DSRL) [29], which steers within the action distribution of $\pi^K_{\text{base}}$ by modifying its initial noise. In contrast, MIDAS applies value-guided residual corrections, allowing executed actions to move beyond high-probability samples from the base policy.

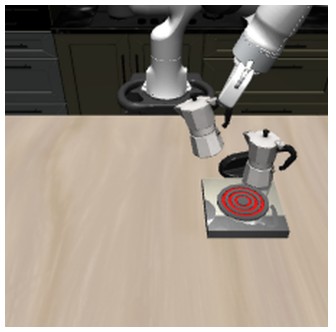 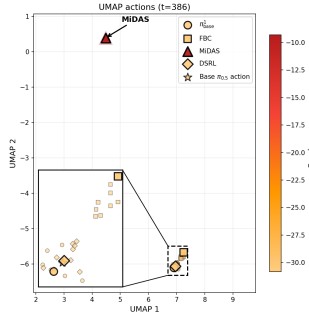

(a) Trajectory critical state     (b) UMAP of actions

**Figure 6: MIDAS reaches actions outside the effective support of $\pi^K_{\text{base}}$.** Filtered BC and DSRL remain near samples from $\pi^K_{\text{base}}$ and fail; MIDAS selects an action in a disjoint region with higher $Q^*$ and succeeds.

Empirically, Filtered BC and DSRL improve only marginally over $\pi^K_{\text{base}}$, while MIDAS achieves substantially higher success Table I. Fig. 6 makes this distinction clear, at a critical state in a LIBERO-Long rollout: actions from Filtered BC and DSRL remain concentrated near samples from $\pi^K_{\text{base}}$ and fail, while MIDAS selects an action in a disjoint region of the action space and succeeds. The fine-grained control missing from $\pi^K_{\text{base}}$ in this case does not lie within the base policy's effective action prior; sharpening or steering within that prior cannot reach it. Closing the control gap, therefore, requires online adaptation that can expand the action distribution beyond the effective support of $\pi^K_{\text{base}}$.

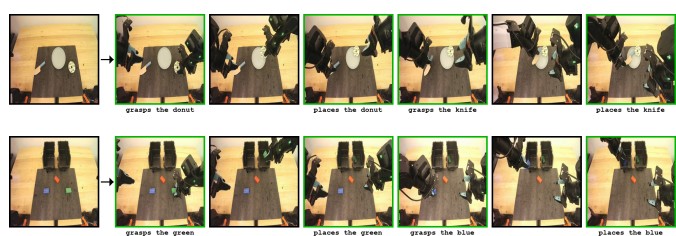

**Figure 7: Real-world rollout comparisons.**

**4) MIDAS transfers to the real world**
We finally evaluate whether minimal-data adaptation is tractable in the real world using a bimanual YAM platform. We consider two pick-and-place tasks: **(T1)** placing a green block in the right container and a blue block in the left container, and **(T2)** placing a knife and a donut on a plate (Figure 7). In both tasks, we use $\pi_{0.5}$ [15] as $\pi_{\text{base}}$ and apply the same MIDAS protocol with a single demonstration ($K=1$); implementation details are deferred to Appendix J. On **T1**, the one-demo policy $\pi^1_{\text{base}}$ already reaches $40\%$ success, which we hypothesize reflects overlap with the pretraining distribution of $\pi_{0.5}$. Online RL further improves success to $67\%$ and produces qualitatively new corrective behaviors not observed in base-policy rollouts (Appendix J). On the harder **T2** task, where the warm-start policy is substantially weaker, online adaptation improves success from $27\%$ to $80\%$ after 5-6 hours of autonomous interaction. These results show that MiDAS

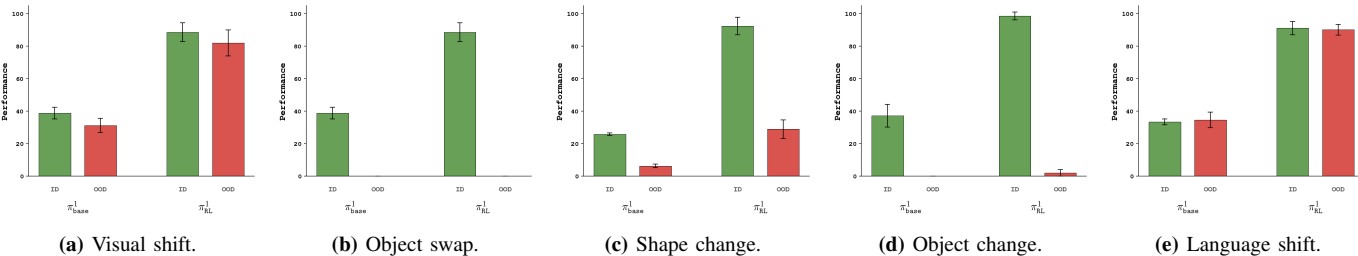

**Figure 5:** Performance of BC and online RL under observation and state shifts. **(a, e)** Observation shifts: pretrained representations preserve performance under visual and language changes. **(b–d)** State shifts: online RL improves robustness to moderate shifts but performance degrades under larger shifts.

can be effective on real hardware, even when the one demo warm-started policy is not very reliable.

## VI. EVALUATING THE GENERALIZATION AND ROBUSTNESS OF MIDAS

Above, we exposed how the pre-trained policy, demonstrations, and online-improvement enable MIDAS to improve performance from the few initial conditions shown in the demonstrations. Here, we ask: can MIDAS generalize beyond the narrow support of initial states, and if so, how?

To test generalization, we consider two categories of shifts: **(a) observation shifts**, which change task-irrelevant inputs such as object appearance (color, texture) or language phrasing (instruction paraphrases), and **(b) state shifts**, which change dynamically relevant quantities such as starting positions, object shapes, or object instances. We ask: how does MIDAS generalize **(1)** to observation shifts in vision and language? **(2)** to shifts in state shifts in position, shape, and object identity? and **(3)**: can autonomous learning alone improve generalization to larger state shifts? As in Section V, we evaluate on the 10 long-horizon tasks in LIBERO-Long. For visual, shape, and object-instance shifts, we use the perturbation definitions from LIBERO-PRO [33]. For position shifts, we perturb target objects at three levels: low (5 cm / 0.1 rad), medium (10 cm / 0.15 rad), and high (20 cm / 0.25 rad). (Further details in Section D-B)

**(Q1) Observation-level robustness is inherited from the frozen vision-language backbone.** We first test whether minimal-data adaptation preserves robustness to semantic input changes, rather than merely fitting the visual and language conditions seen in the demonstrations. Under visual shifts such as color and texture changes (Fig. 5a), $\pi_{\text{base}}^{K}$ ($K = 1$) retains most of its in-distribution performance, and the adapted policy $\pi_{\text{RL}}^{1}$ shows a similar retention pattern despite using only a lightweight residual actor. The same holds under language paraphrases (Fig. 5e). Since both policies share the same frozen VLM backbone, and the RL module adds little perceptual or linguistic capacity, this robustness is best explained by invariances already present in the pretrained representation. Thus, online RL improves control without sacrificing observation-level generalization.

**(Q2) Generalization to state shifts depends on whether new behavior is required.** We next evaluate shifts that require the policy to change its actions. Figures 5c, 5b, and 5d show that $\pi_{\text{RL}}^{1}$ improves over $\pi_{\text{base}}^{1}$ under some state shifts, especially shape changes where the required grasp is similar. This

makes sense: the policy must execute the same behavior in a different part of the state space. However, performance under both drops sharply under object swaps and object changes: changes in object affordance or placement can require distinct manipulation strategies, which are difficult to infer from one demonstration and online interaction around it. (Further details in Section D-C).

**(Q3) Broader coverage, either from demonstrations or curriculum, expands the generalization boundary.** Lastly, we ask: can autonomous adaptation ensure robustness to large perturbation shifts that are outside the support of the reset distribution, provided that they do not require fundamentally new behavior (in view of Q2). In the Appendix Section H, we show that naively increasing the breadth of demonstration can improve robustness. In Appendix Section G, we describe a curriculum that gradually increases the width of the distribution, and achieves similar robustness to a policy trained on 50 demonstrations.

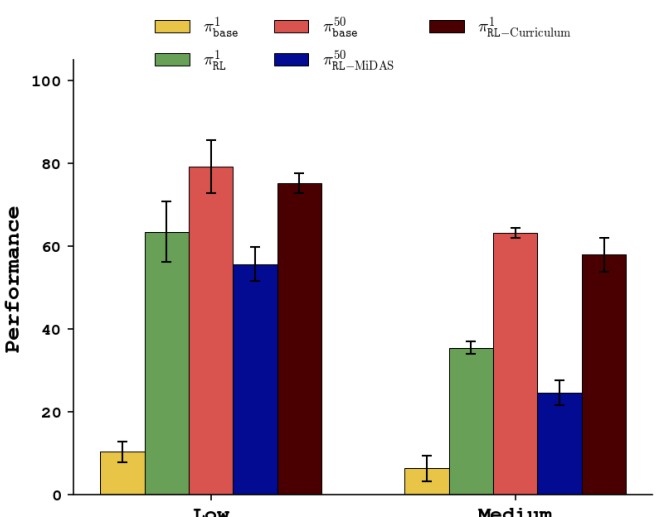

**Figure 8: Curriculum closes the gap.** Progressively widening the reset distribution during online RL recovers most of the position-generalization gap with $\pi_{\text{base}}^{50}$ without additional demos.

## VII. LIMITATIONS

The results in this paper should be interpreted within two limitations. First, the present instantiation remains limited on longer-horizon tasks, where sparse-reward value learning must assign credit through long sequences of actions proposed by

a frozen prior. Scaling MDA may therefore require stronger value priors, such as pretrained critics or value decompositions that expose intermediate progress signals. Second, Stage 2 remains reset-dependent and interaction-intensive. This is manageable in simulation, but becomes a central constraint on hardware, where failures, resets, and safety limit autonomous practice without an operator in the loop. Reducing this dependence will require safer exploration, better failure detection, learned resets, or lightweight intervention mechanisms that preserve the low-demonstration setting. We discuss directions for future work in Appendix K.

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

Additional real-world videos and demonstration rollouts are available on the project website: coruscating-pika-920eda.netlify.app.

## APPENDIX A
## ACTION CHUNKING

We apply action chunking [32], where sequences of actions $u_{t:t+C-1}$ are predicted and executed in open-loop. Let $u_t \in \mathbb{R}^d$ denote a primitive robot action and let $C$ denote the chunk horizon. We define the action space used by the policy as $\mathcal{A} \subseteq \mathbb{R}^{C \times d}$, so that each action $a_t \in \mathcal{A}$ corresponds to an entire action chunk, $a_t = (u_t, \ldots, u_{t+C-1})$. After sampling $a_t \sim \pi(\cdot \mid s_t)$, the robot executes the chunk open-loop for $C$ environment steps before replanning from the resulting state $s_{t+C}$. For notational simplicity, we treat each chunk as a single action in the MDP, preserving the standard MDP notation above.

When using chunked actions, the reward associated with a transition is the discounted reward accumulated over the executed chunk, and bootstrapping occurs from $s_{t+C}$ with discount $\gamma^C$. Thus, unless stated otherwise, $a_t$ denotes an

action chunk and $r_t$ denotes the reward accumulated over the executed chunk, with bootstrapping discount $\gamma^C$.

## APPENDIX B
## RELATED WORK

*a) Generalist robot policies and VLAs.:* Large-scale robot pretraining has enabled generalist policies that can condition on language and visual observations to perform diverse manipulation tasks. Prior work has introduced transformer-based robot policies and vision-language-action models trained on broad robot and web-scale data, including RT-1 [6], RT-2 [5], Octo [27], $\pi_0$ [4], $\pi_{0.5}$ [15], world-action models [31], and efficient VLA fine-tuning methods [16]. These models provide reusable perceptual representations and broad action priors, making them natural candidates for fast downstream adaptation. However, despite this progress, current generalist policies are still not reliable zero-shot learners in new environments: they may fail to reach sparse rewards or visit too narrow a set of task-relevant states for autonomous learning to bootstrap. In contrast, we study whether such pretrained policies can be made adaptable from minimal task-specific supervision.

*b) Few-shot robot imitation and adaptation.:* A complementary line of work studies learning from very small numbers of demonstrations. One-shot and few-shot imitation learning methods [8, 9] show that robots can infer new tasks from one or a few examples, often through meta-learning or in-context task conditioning [24]. Other approaches study rapid adaptation from short demonstrations [12] or in-context imitation policies [28]. These methods show that limited demonstrations can induce task-solving behavior, but they generally aim to solve the task through imitation or test-time conditioning alone, and do not focus on subsequent autonomous improvement or the analysis of what contributes to this improvement.

*c) Online and residual RL for robot policy adaptation.:* Online reinforcement learning and offline-to-online methods provide tools for improving robot policies after deployment. Prior work has studied offline RL pretraining [18], calibrated offline-to-online fine-tuning [22], efficient online RL with offline data [3], imitation-bootstrapped RL [14], and policy-agnostic RL for fine-tuning diverse policy classes [21]. More closely related, recent methods use RL to improve expressive policies or pretrained robot policies, including diffusion-policy steering [29], diffusion policy optimization [25], RL-token adaptation [30], distribution-contractive fine-tuning [26], and residual RL for refining imitation policies [7, 1, 2]. These methods show that value-guided online improvement can refine pretrained or imitation-learned policies, given access to 10-100 expert demonstrations. MIDAS builds on this direction, but focuses on the minimal-data setting where only one or a few demonstrations are available. Rather than assuming broad offline data or a strong initial policy, MIDAS first uses few-demo behavior cloning to make task-relevant behavior reachable, then freezes the pretrained backbone and trains a lightweight residual actor with value-based RL to learn

corrective actions beyond the effective support of the few-demo policy.

## APPENDIX C
## RESIDUAL POLICY PARAMETERIZATION

For the state representation $s_t$, we use the output of the VLM head by taking a mean across all token positions. The residual actor is parameterized as a 3-layer MLP with a hidden dimension of 256 and ReLU activation.

## APPENDIX D
## SIMULATION EXPERIMENT DETAILS

We use the Libero and Robocasa Benchmarks for our simulation experiments

### A. Libero and Robocasa Task Details

We evaluate on two simulation benchmarks: LIBERO-Long [19] and RoboCasa-365 [23]. Below we describe each benchmark and the specific tasks used.

*a) LIBERO-Long.:* LIBERO-Long consists of 10 long-horizon, language-conditioned manipulation tasks performed by a fixed-base Franka Panda arm. Each task requires completing multiple subtasks in sequence (e.g., picking two objects and placing them in a container). The 10 tasks are:

1) Put both the alphabet soup and the tomato sauce in the basket
2) Put both the cream cheese box and the butter in the basket
3) Turn on the stove and put the moka pot on it
4) Put the black bowl in the bottom drawer of the cabinet and close it
5) Put the white mug on the left plate and put the yellow and white mug on the right plate
6) Pick up the book and place it in the back compartment of the caddy
7) Put the white mug on the plate and put the chocolate pudding to the right of the plate
8) Put both the alphabet soup and the cream cheese box in the basket
9) Put both moka pots on the stove
10) Put the yellow and white mug in the microwave and close it

Each task has 50 human-teleoperated demonstrations available in the benchmark; we use only $K = 1$ for adaptation in our main experiments. The environment provides two camera views: an `agentview` third-person camera and a `robot0_eye_in_hand` wrist camera, both rendered at $224 \times 224$ pixels. The action space is 7-dimensional: 3D end-effector position, 3D end-effector rotation (axis-angle), and 1D gripper open/close. The episode horizon is 500 timesteps. The learning curves for all the tasks can be visualized in Fig. 9.

*b) RoboCasa-365.:* RoboCasa-365 [23] provides long-horizon household manipulation tasks in diverse procedurally generated kitchen scenes. Unlike LIBERO, the robot is a mobile manipulator and tasks involve navigation, object retrieval, and multi-step interaction with kitchen appliances. We evaluate on four tasks:

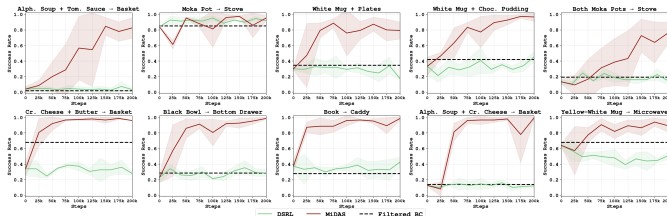

**Figure 9:** Libero-Long Learning Curves for MIDAS with baseline comparison. Solid line shows mean across 3 seeds and shaded region denotes standard deviation.

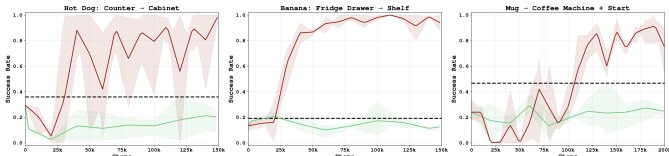

**Figure 10:** Robocasa-365 Learning Curves for MIDAS with baseline comparison. Solid line shows mean across 3 seeds and shaded region denotes standard deviation.

1) **Banana: Fridge Drawer → Shelf** (`PickPlaceFridgeDrawerToShelf`): Pick a banana from a fridge drawer and place it on a shelf.
2) **Hot Dog: Counter → Cabinet** (`PickPlaceCounterToCabinet`): Pick a hot dog from the counter and place it in a cabinet.
3) **Mug → Coffee Machine + Start** (`PrepareCoffee`): Place a mug under the coffee machine and press the start button.
4) **Cup + Bowl → Dishwasher + Close** (`LoadDishwasher`): Load a cup and bowl into the dishwasher and close it.

Each task is instantiated in a specific kitchen layout and style that is fixed across training and evaluation. The environment provides two camera views: `robot0_agentview_left` and `robot0_eye_in_hand`, rendered at $256 \times 256$ pixels. The action space is 7-dimensional for the first three tasks (3D end-effector position, 3D rotation, 1D gripper), with episode horizons varying between 500 and 800 timesteps depending on task complexity (except for `LoadDishwashwer`). The learning curves for all the tasks can be visualized in Fig. 10.

**Long Horizon Manipulation:** Note that the `LoadDishwasher` task spans a horizon of 1100 timesteps and has an overall success rate of 0% across all baselines and MIDAS. This task shows the limitation of our proposed approach and provides motivation for future work to extend MIDAS to long-horizon, sparse-reward settings.

*c) Shared observation and action details.:* In both benchmarks, the base policy ($\pi_{0.5}$) receives $224 \times 224$ images after resizing with padding. Actions are temporally chunked: the policy outputs a chunk of $C = 10$ primitive actions that are executed open-loop before replanning (see Section A). The residual learner receives images resized to $100 \times 100$ for computational efficiency, along with the base policy's proposed action chunk. Evaluation uses 50 rollouts per task from the canonical initial state distribution.

## B. Libero Pro Perturbation Details

We use the perturbation protocol from LIBERO-PRO to evaluate generalization under controlled distribution shifts. LIBERO-PRO defines five perturbation axes, of which we use four (excluding task perturbation, which changes the goal itself). We also introduce position perturbations at three magnitudes to test spatial generalization. Not all perturbation types apply to all 10 tasks; LIBERO-PRO defines perturbations only for tasks where a given shift is meaningful. Below we describe each perturbation type along with the specific tasks used.

*a) Observation-level perturbations.:* These change task-irrelevant aspects of the observation while preserving the underlying state and goal.

- **Visual (object appearance)** (Fig. 5a): Object colors, textures, and sizes are modified in the scene definition (e.g., an `akita_black_bowl` is replaced with a red or larger variant), while the task instruction and goal conditions remain unchanged. We evaluate on 5 tasks where LIBERO-PRO defines color or texture swaps that do not alter the object's shape or grasp affordance: *Both Moka Pots → Stove*, *Cr. Cheese + Butter → Basket*, *Moka Pot → Stove*, *White Mug + Choc. Pudding*, *White Mug + Plates*.
- **Language** (Fig. 5e): The task instruction is replaced with a paraphrase that describes the same goal using different phrasing (e.g., "put both moka pots on the stove" → "place both moka pots on stove"). Object references and goal logic are preserved. We evaluate on all 10 LIBERO-Long tasks, as language paraphrases are defined for every task.

*b) State-level perturbations.:* These change dynamically relevant quantities that may require different actions. LIBERO-PRO applies a single object-replacement perturbation per task, but the nature of the replacement varies: for some tasks the replacement changes only the object's shape while preserving its category, for others it substitutes a different object category entirely, and for others the two scene objects swap positions. We group results by the type of change each replacement induces.

- **Object swap** (Fig. 5b): The initial positions of two objects are exchanged by swapping their spawn-region assignments in the scene definition. The policy must execute the task with objects in different spatial locations. We evaluate on the same 5 tasks as the visual perturbation: *Both Moka Pots → Stove*, *Cr. Cheese + Butter → Basket*, *Moka Pot → Stove*, *White Mug + Choc. Pudding*, *White Mug + Plates*.
- **Shape change** (Fig. 5c): Target object meshes are replaced with geometrically different variants of the same category (e.g., a taller mug or wider bowl). If the new shape preserves similar grasp affordances, the required manipulation strategy remains the same; otherwise the policy must adapt its contact behavior. We evaluate on 4 tasks where the LIBERO-PRO replacement alters the

object geometry: *Black Bowl → Bottom Drawer, Yellow-White Mug → Microwave, Alph. Soup + Cr. Cheese → Basket, Alph. Soup + Tom. Sauce → Basket*.

- **Object change** (Fig. 5d): The target object is replaced with a different object category entirely, changing both visual appearance and manipulation affordances. We evaluate on 1 task where the replacement introduces a categorically different object: *Book → Caddy*.

*c) Position perturbation.:* We evaluate position generalization by displacing target objects from their canonical initial positions at three levels:

- **Low:** 5 cm XY displacement, 0.1 radians yaw perturbation
- **Medium:** 10 cm XY displacement, 0.15 radians yaw perturbation
- **High:** 20 cm XY displacement, 0.25 radians yaw perturbation

XY offsets are sampled uniformly within a disk of the specified radius. After applying perturbations, objects are settled under physics simulation and validated: the perturbed object must remain visible in the camera frame, stay within 2 cm of its settled z-height (no falling off surfaces), not collide with other scene objects, and maintain an upright orientation (local up-axis within $18°$ of the world vertical). Invalid samples are re-drawn for up to 50 attempts per episode.

We evaluate position perturbation on 3 tasks: *Alph. Soup + Tom. Sauce → Basket, Black Bowl → Bottom Drawer, Both Moka Pots → Stove*. Perturbations are applied to the task-relevant target objects in each case (e.g., both moka pots, or the black bowl).

*d) Evaluation protocol.:* All perturbation evaluations use 50 rollouts per task. For LIBERO-PRO perturbations (visual, language, swap, shape, object change), we evaluate from the perturbed initial configurations using the standard episode horizon. For position perturbations, we evaluate at each magnitude separately. Results are averaged across 3 seeds. Per-task breakdowns are provided in the tables below.

*e) Visual perturbation results.:* Table II reports per-task success rates under visual (object appearance) perturbation.

| Task | $\pi^1_{\text{base}}$ | | $\pi^1_{\text{RL}}$ | |
|---|---|---|---|---|
| | ID | OOD | ID | OOD |
| Both Moka Pots → Stove | 13.3±3.1 | 6.0±2.0 | 76.0±14.0 | 58.7±33.5 |
| Cr. Cheese + Butter → Basket | 34.0±9.2 | 38.0±9.2 | 96.0±0.0 | 98.7±2.3 |
| Moka Pot → Stove | 84.0±0.0 | 58.7±5.0 | 95.3±3.1 | 93.3±4.2 |
| White Mug + Choc. Pudding | 33.3±11.7 | 30.0±10.4 | 96.7±5.8 | 80.0±11.1 |
| White Mug + Plates | 30.0±2.0 | 24.0±2.0 | 79.3±11.0 | 79.3±13.6 |
| **Average** | 38.9±3.6 | 31.3±4.4 | 88.7±5.9 | 82.0±8.0 |

**Table II:** Per-task success rate (%) under visual (object appearance) perturbation. OOD replaces object colors, textures, or sizes while preserving shape and task goal.

*f) Language perturbation results.:* Table III reports per-task success rates under language perturbation.

## C. Libero Pro State Shift Perturbation Experiments

Here we provide further analysis of the state-shift generalization results from Section V.

| Task | $\pi^1_{\text{base}}$ | | $\pi^1_{\text{RL}}$ | |
|---|---|---|---|---|
| | ID | OOD | ID | OOD |
| Alph. Soup + Cr. Cheese → Basket | 12.7±1.2 | 14.0±2.0 | 99.3±1.2 | 98.0±2.0 |
| Alph. Soup + Tom. Sauce → Basket | 4.0±3.5 | 4.7±3.1 | 82.7±13.3 | 78.7±11.0 |
| Black Bowl → Bottom Drawer | 22.7±6.4 | 23.3±9.2 | 98.7±1.2 | 96.7±5.8 |
| Book → Caddy | 37.3±7.0 | 33.3±8.3 | 98.7±2.3 | 96.7±4.2 |
| Both Moka Pots → Stove | 13.3±3.1 | 15.3±2.3 | 76.0±14.0 | 79.3±13.0 |
| Cr. Cheese + Butter → Basket | 34.0±9.2 | 40.0±14.0 | 96.0±0.0 | 96.0±2.0 |
| Moka Pot → Stove | 84.0±0.0 | 88.7±5.8 | 95.3±3.1 | 97.3±2.3 |
| White Mug + Choc. Pudding | 33.3±11.7 | 38.7±13.3 | 96.7±5.8 | 91.3±8.1 |
| White Mug + Plates | 30.0±2.0 | 33.3±5.8 | 79.3±11.0 | 78.0±2.0 |
| Yellow-White Mug → Microwave | 64.0±2.0 | 56.0±6.9 | 89.3±8.1 | 90.0±7.2 |
| **Average** | 33.5±1.8 | 34.7±4.8 | 91.2±4.2 | 90.2±3.3 |

**Table III:** Per-task success rate (%) under language perturbation. OOD replaces the task instruction with a paraphrase.

*a) Object swap.:* When two objects exchange positions, the policy must approach the target from a different direction or reorder its subtask sequence. On the 5 evaluated tasks (*Both Moka Pots → Stove, Cr. Cheese + Butter → Basket, Moka Pot → Stove, White Mug + Choc. Pudding, White Mug + Plates*), swapping object positions changes the optimal grasp order and approach trajectory. Both $\pi^1_{\text{base}}$ and $\pi^1_{\text{RL}}$ achieve 0% success across all 5 tasks under swap perturbation (Table IV), despite strong ID performance. This indicates that swapping object positions produces configurations far enough from the training distribution that neither the BC warm start nor residual RL corrections can recover.

| Task | $\pi^1_{\text{base}}$ | | $\pi^1_{\text{RL}}$ | |
|---|---|---|---|---|
| | ID | OOD | ID | OOD |
| Both Moka Pots → Stove | 13.3±3.1 | 0.0±0.0 | 76.0±14.0 | 0.0±0.0 |
| Cr. Cheese + Butter → Basket | 34.0±9.2 | 0.0±0.0 | 96.0±0.0 | 0.0±0.0 |
| Moka Pot → Stove | 84.0±0.0 | 0.0±0.0 | 95.3±3.1 | 0.0±0.0 |
| White Mug + Choc. Pudding | 33.3±11.7 | 0.0±0.0 | 96.7±5.8 | 0.0±0.0 |
| White Mug + Plates | 30.0±2.0 | 0.0±0.0 | 79.3±11.0 | 0.0±0.0 |
| **Average** | 38.9±3.6 | 0.0±0.0 | 88.7±5.9 | 0.0±0.0 |

**Table IV:** Per-task success rate (%) under object swap perturbation. OOD exchanges the initial positions of two scene objects.

*b) Shape change.:* Shape perturbations test whether learned grasps transfer across object geometries. On the 4 evaluated tasks (*Black Bowl → Bottom Drawer, Yellow-White Mug → Microwave, Alph. Soup + Cr. Cheese → Basket, Alph. Soup + Tom. Sauce → Basket*), the replacement preserves the object category but alters its geometry. Table V shows that $\pi^1_{\text{RL}}$ retains strong performance on tasks where the new shape preserves similar grasp affordances (e.g., Black Bowl at 90.7%), but drops sharply when the shape change alters the required contact geometry (e.g., Alph. Soup tasks at <3%).

| Task | $\pi^1_{\text{base}}$ | | $\pi^1_{\text{RL}}$ | |
|---|---|---|---|---|
| | ID | OOD | ID | OOD |
| Black Bowl → Bottom Drawer | 22.7±6.4 | 20.7±6.4 | 98.7±1.2 | 90.7±8.1 |
| Yellow-White Mug → Microwave | 64.0±2.0 | 4.7±3.1 | 89.3±8.1 | 20.7±13.3 |
| Alph. Soup + Cr. Cheese → Basket | 12.7±1.2 | 0.0±0.0 | 99.3±1.2 | 2.7±4.6 |
| Alph. Soup + Tom. Sauce → Basket | 4.0±3.5 | 0.0±0.0 | 82.7±13.3 | 2.0±3.5 |
| **Average** | 25.8±0.8 | 6.3±1.0 | 92.5±5.4 | 29.0±5.8 |

**Table V:** Per-task success rate (%) under shape change perturbation. OOD replaces the target object with a geometrically different variant of the same category.

*c) Object change.:* Replacing the target with a different object category represents the most severe state shift. We evaluate this on *Book → Caddy*, where the book is replaced with a categorically different object. Table VI shows that both $\pi_{\text{base}}^1$ and $\pi_{\text{RL}}^1$ drop to near 0% under this shift despite strong ID performance. The new object differs in size, shape, weight, and affordance, requiring manipulation strategies outside the behavioral support of $\pi_{\text{RL}}^1$. This category of shift cannot be addressed by online RL alone and requires additional demonstrations or interaction with the new object.

| Task | $\pi_{\text{base}}^1$ | | $\pi_{\text{RL}}^1$ | |
|---|---|---|---|---|
| | ID | OOD | ID | OOD |
| Book → Caddy | 37.3±7.0 | 0.0±0.0 | 98.7±2.3 | 2.0±2.0 |

**Table VI:** Per-task success rate (%) under object change perturbation. OOD replaces the target with a categorically different object.

*d) Position perturbation.:* Tables VII,VIII,IX reports per-task success rates under position perturbation at three displacement levels. We compare $\pi_{\text{base}}^1$, $\pi_{\text{RL}}^1$, and $\pi_{\text{base}}^{50}$ (behavior-cloned on all 50 demonstrations). $\pi_{\text{RL}}^1$ improves over $\pi_{\text{base}}^1$ at all levels but falls short of $\pi_{\text{base}}^{50}$, especially at larger displacements where the gap in state coverage becomes the binding constraint.

| Task | $\pi_{\text{base}}^1$ | $\pi_{\text{RL}}^1$ | $\pi_{\text{base}}^{50}$ |
|---|---|---|---|
| Alph. Soup + Tom. Sauce | 4.0±2.0 | 52.7±5.8 | 94.7±5.8 |
| Black Bowl → Drawer | 24.0±7.2 | 88.7±5.0 | 59.3±10.1 |
| Both Moka Pots → Stove | 3.3±3.1 | 49.3±20.1 | 84.0±7.2 |
| **Average** | 10.4±2.5 | 63.6±7.3 | 79.3±6.4 |

**Table VII:** Per-task success rate (%) under low position perturbation of 5 cm.

| Task | $\pi_{\text{base}}^1$ | $\pi_{\text{RL}}^1$ | $\pi_{\text{base}}^{50}$ |
|---|---|---|---|
| Alph. Soup + Tom. Sauce | 1.3±2.3 | 14.0±2.0 | 68.7±3.1 |
| Black Bowl → Drawer | 16.7±6.4 | 70.0±3.5 | 45.3±2.3 |
| Both Moka Pots → Stove | 1.3±1.2 | 22.7±8.3 | 76.0±7.2 |
| **Average** | 6.4±3.1 | 35.6±1.5 | 63.3±1.2 |

**Table VIII:** Per-task success rate (%) under medium position perturbation of 10 cm.

| Task | $\pi_{\text{base}}^1$ | $\pi_{\text{RL}}^1$ | $\pi_{\text{base}}^{50}$ |
|---|---|---|---|
| Alph. Soup + Tom. Sauce | 0.0±0.0 | 2.7±3.1 | 25.3±4.2 |
| Black Bowl → Drawer | 6.7±2.3 | 37.3±2.3 | 23.3±8.1 |
| Both Moka Pots → Stove | 0.0±0.0 | 2.7±3.1 | 25.3±9.5 |
| **Average** | 2.2±0.8 | 14.2±0.4 | 24.7±4.4 |

**Table IX:** Per-task success rate (%) under high position perturbation of 20 cm.

# APPENDIX E
## IMPLEMENTATION DETAILS

### A. PARL

**State representation.** The state input to both the residual actor and critic is derived from the frozen VLA backbone of $\pi_{\text{base}}^K$. At each timestep, we extract the VLM token embeddings and mean-pool across all token positions to produce a 2048-dimensional feature vector. This vector is then projected through a bottleneck layer (linear projection, layer normalization, and tanh activation) to yield a 200-dimensional latent state $\mathbf{s}_t$.

**Actor inputs and architecture.** The residual actor receives the bottlenecked VLM embedding concatenated with the flattened base action chunk $\mathbf{a}_t^{\text{base}} \in \mathbb{R}^{C \cdot d_a}$ proposed by $\pi_{\text{base}}^K$, where $C=10$ is the chunk length and $d_a=7$ is the per-step action dimension. As described in Section IV, the actor directly outputs the full executed action chunk $\mathbf{a}_t^{\text{exec}}$ rather than an additive residual correction. The actor is parameterized as a 3-layer MLP with 512 hidden units per layer and ReLU activations. The output distribution is a tanh-squashed diagonal Gaussian with learned mean and log-standard-deviation heads.

**Critic inputs and architecture.** The critic receives only the bottlenecked VLM embedding as its state input; the base action chunk is excluded. Given a candidate executed action $\mathbf{a}_t^{\text{exec}}$, each Q-network in the ensemble evaluates $Q_{\psi_j}(\mathbf{s}_t, \mathbf{a}_t^{\text{exec}})$. We use an ensemble of 10 Q-networks, each parameterized as a 3-layer MLP with 512 hidden units and ReLU activations. The ensemble aggregates predictions via mean reduction during target computation.

**PA-RL procedure.** At each actor update, a single base action $\mathbf{a}_t^{\text{base}}$ is sampled from the frozen $\pi_{\text{base}}^K$. The residual actor then proposes $N=16$ candidate actions conditioned on $\mathbf{s}_t$ and $\mathbf{a}_t^{\text{base}}$. The top $M=8$ actions ranked by the critic ensemble are selected as elites. These elite actions undergo 30 steps of gradient ascent on the Q-ensemble with step size $10^{-3}$. The actor is then updated via a distillation loss that regresses toward the refined elite actions.

**Optimization.** Both actor and critic use the Adam optimizer with global gradient norm clipping at 1.0. The actor learning rate is $10^{-4}$ and the critic learning rate is $3 \times 10^{-4}$. At each environment step, we perform 30 critic updates and 10 actor updates. Training begins with a BC warmup phase of 10,000 steps during which the residual actor is trained to reproduce the base policy proposals and the critic is calibrated on the warmup buffer, each with 8 updates per step.

**Key hyperparameters.** Table X summarizes the hyperparameters used for all PARL training runs.

### B. BC Implementation

For behavior cloning, we use a single demonstration out of the available 50 for LIBERO-Long and one demonstration for the particular scene and layout configuration for each RoboCasa task. We then fine-tune $\pi_{0.5}$ using the official OpenPI codebase on this demonstration for 16,000 gradient steps. Table XII shows the performance on of behavior cloning on varying the number of demos $K = \{1, 3, 5\}$.

**Table X:** PARL hyperparameters used for all training runs.

| Hyperparameter | Value |
|---|---|
| Batch size | 64 |
| Discount factor ($\gamma$) | 0.999 |
| Target network update rate ($\tau$) | 0.05 |
| Actor learning rate | $10^{-4}$ |
| Critic learning rate | $3 \times 10^{-4}$ |
| Hidden dimensions | $(512, 512, 512)$ |
| VLM embedding bottleneck dim | 200 |
| Number of Q-networks | 10 |
| Critic ensemble reduction | mean |
| PA-RL candidate samples ($N$) | 16 |
| PA-RL elite samples ($M$) | 8 |
| PA-RL gradient steps | 30 |
| PA-RL step size | $10^{-3}$ |
| Critic updates per env step | 30 |
| Actor updates per env step | 10 |
| BC warmup steps | 10,000 |
| BC warmup critic/actor updates | 8 |
| Success buffer ratio ($\rho_{\mathrm{succ}}$) | 0.20 |
| Success buffer minimum size | 20 |
| Gradient norm clip | 1.0 |
| Optimizer | Adam |
| Action chunk length ($C$) | 10 |

**Table XI:** Stage I behavior cloning hyperparameters.

| Hyperparameter | Value |
|---|---|
| Number of demonstrations ($K$) | 1 |
| Training steps | 16,000 |
| Batch size | 32 |
| Optimizer | AdamW |
| Peak learning rate | $2.5 \times 10^{-5}$ |
| LR schedule | Cosine decay |
| LR warmup steps | 50 |
| Minimum learning rate | $2.5 \times 10^{-6}$ |
| $\beta_1, \beta_2$ | 0.9, 0.95 |
| Gradient norm clip | 1.0 |
| LoRA rank | 32 |
| LoRA $\alpha$ | 32 |
| LoRA targets | Attention + FFN |
| Action expert | Fully finetuned |
| Vision encoder (SigLIP) | Fully finetuned |
| Action horizon ($C$) | 10 |
| Loss | Flow matching |

| | BC Success Rate (%) | | |
|---|---|---|---|
| Task | $K=1$ | $K=3$ | $K=5$ |
| *LIBERO-Long* | | | |
| Alphabet Soup + Cream Cheese $\to$ Basket | 12.7 | 39.3 | **82.7** |
| Alphabet Soup + Tomato Sauce $\to$ Basket | 4.0 | 32.7 | **74.7** |
| Black Bowl $\to$ Bottom Drawer | 22.7 | 90.0 | **94.0** |
| Book $\to$ Caddy | 37.3 | 26.7 | **89.3** |
| Both Moka Pots $\to$ Stove | 13.3 | **51.3** | 35.3 |
| Cream Cheese + Butter $\to$ Basket | 34.0 | **90.7** | 84.7 |
| Moka Pot $\to$ Stove | 84.0 | 86.0 | **94.7** |
| White Mug + Chocolate Pudding | 33.3 | 26.7 | **66.0** |
| White Mug + Plates | 30.0 | 48.7 | **74.0** |
| Yellow-White Mug $\to$ Microwave | 64.0 | 70.7 | **86.0** |
| **Average** | 33.5 | 56.3 | **78.1** |

**Table XII: Per-task BC success on LIBERO-Long.** Bold denotes the best result per task across $K \in \{1, 3, 5\}$ demonstrations.

**Trainable parameters.** We apply LoRA adapters to the PaliGemma 2B vision-language backbone, inserting low-rank updates into both the attention and feed-forward layers with rank 32 and scaling factor $\alpha=32$. The 300M-parameter action expert (Gemma 300M) and the SigLIP vision encoder are fully finetuned. All other backbone weights remain frozen. This selective finetuning keeps the majority of the VLM fixed while allowing the action head and vision encoder to specialize to the target task.

**Loss and optimization.** The model is trained with the flow-matching objective native to the $\pi_{0.5}$ architecture, predicting action chunks of horizon $C=10$. We use AdamW with $\beta_1=0.9$, $\beta_2=0.95$, and negligible weight decay ($10^{-10}$), with global gradient norm clipping at 1.0. The learning rate follows a cosine decay schedule: peak learning rate $2.5 \times 10^{-5}$, 50 warmup steps, decaying to $2.5 \times 10^{-6}$ over 100,000 steps. Training is stopped at 16,000 steps. The batch size is 32.

**Key hyperparameters.** Table XI lists the hyperparameters used for all Stage I behavior cloning runs.

### C. Success Buffer

The success buffer $\mathcal{B}_{\mathrm{succ}}$ is a separate replay buffer that stores only transitions from successful episodes. Its purpose is to prevent actor updates from being dominated by failed rollouts when the policy's success rate is low (2–5% early in training).

**Initialization.** Both the main replay buffer $\mathcal{B}$ and the success buffer $\mathcal{B}_{\mathrm{succ}}$ are initialized with all transitions from the $K$ demonstrations, since demonstration trajectories are successful by construction. The success buffer capacity is set to $\max(10{,}000, 0.5 \times |\mathcal{B}|_{\mathrm{max}})$.

**Population during training.** After each collected trajectory, all transitions are inserted into $\mathcal{B}$. If the episode was successful (sparse reward received), the same transitions are additionally inserted into $\mathcal{B}_{\mathrm{succ}}$.

**Sampling.** Actor update minibatches are sampled from a mixture of the two buffers:

$$\mathcal{B}_{\mathrm{actor}} \sim (1 - \rho_{\mathrm{succ}}) \cdot \mathcal{B} + \rho_{\mathrm{succ}} \cdot \mathcal{B}_{\mathrm{succ}}, \qquad \text{(E.1)}$$

where $\rho_{\mathrm{succ}} = 0.2$ is the success buffer ratio. Concretely, each actor batch of size 64 contains 51 transitions sampled uniformly from $\mathcal{B}$ and 13 transitions sampled uniformly from $\mathcal{B}_{\mathrm{succ}}$. This oversampling biases the actor toward states where successful behavior has been observed, steering the PA-RL elite selection and gradient ascent toward reward-bearing regions.

### APPENDIX F
### ABLATIONS

*a) Design-decision ablations.:* Fig. 11 ablates three implementation choices in MIDAS on the `Both Moka Pots -> Stove` task. Each row changes a single knob while keeping the same PA-RL objective, VLM state representation, 10k-step BC warmup, and one-demonstration initialization.

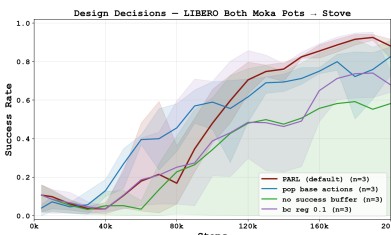

**Figure 11:** MiDAS ablations on both-mokapots task

First, removing the base-policy action from the residual actor nearly matches the default recipe. This suggests that the action head is most important for producing task-relevant warmup experience; once this experience is available, online value learning from pretrained VLA features can recover much of the final performance without using the base action as an explicit actor input. Second, disabling the success buffer tests whether sparse successful transitions must be oversampled during actor updates, rather than being learned from uniformly mixed online replay. Third, adding a persistent success-only BC penalty tests whether the residual should remain behaviorally anchored after warmup, or whether continued RL updates should be allowed to move away from the demonstrated action distribution.

## APPENDIX G
## CURRICULUM ENABLES STRONGER POSITION GENERALIZATION

Table VII, Table VIII, Table IX and Fig. 12b show that $\pi_{\mathrm{RL}}^1$ generalizes well to small position perturbations: at 5 cm displacement, average success reaches 63.8%, compared to 10.4% for $\pi_{\mathrm{base}}^1$. Performance drops sharply at larger displacements, falling to 37.6% at 10 cm and 12.7% at 20 cm. By contrast, $\pi_{\mathrm{base}}^{50}$ maintains 79.3%, 63.3%, and 24.7% at the same levels, indicating that broader training coverage sustains robustness where $\pi_{\mathrm{RL}}^1$ does not.

A key observation is that larger position perturbations in our setting can be decomposed incrementally: a 10 cm displacement is reachable through a sequence of smaller shifts from the nominal configuration. If the policy can reliably solve the task under 5 cm perturbation, it should be able to extend to 10 cm once it has practiced under the smaller shift. This motivates a curriculum over the reset distribution: rather than training under the full perturbation range from the outset, we progressively widen the displacement radius as the policy masters each level.

**Curriculum procedure.** Starting from the trained $\pi_{\mathrm{RL}}^1$ checkpoint (after Stage II of MiDAS ), we continue online RL under position-perturbed resets. The curriculum proceeds through three stages with increasing perturbation radii: 5 cm, 10 cm, and 20 cm. At each stage, the target objects are displaced uniformly at random within the specified radius from their default positions. The policy advances to the next stage once its evaluation success rate on the current perturbation level exceeds a threshold of 0.6. On the final stage, training

continues until the step budget is exhausted. All other hyperparameters (critic/actor updates, learning rates, success buffer) remain identical to the standard MiDAS Stage II configuration (Table X). No additional demonstrations are provided; the curriculum relies entirely on autonomous interaction.

**Task selection.** We run the curriculum on the three tasks used for position perturbation evaluation: *Alphabet Soup + Tomato Sauce → Basket*, *Both Moka Pots → Stove*, and *Black Bowl → Bottom Drawer*. For *Black Bowl*, $\pi_{\mathrm{RL}}^1$ already achieves 88.7% at 5 cm and 70.0% at 10 cm without curriculum (Table VIII), both exceeding the 0.6 advancement threshold. We therefore exclude it from the curriculum experiment, as it already demonstrates strong position generalization under the standard MiDAS recipe.

**Results.** Table XIII reports per-task success rates after curriculum training, compared against the non-curriculum $\pi_{\mathrm{RL}}^1$ and $\pi_{\mathrm{base}}^{50}$ baselines from Table VII, Table VIII, Table IX. On *Both Moka Pots*, curriculum raises 5 cm success from 49.3% to 70.0% and 10 cm success from 22.7% to 75.3%, nearly matching $\pi_{\mathrm{base}}^{50}$ (76.0%) at the medium perturbation level. On *Alphabet Soup + Tomato Sauce*, 5 cm success improves from 52.7% to 68.7% and 10 cm success from 14.0% to 28.7%. The gains are consistent across seeds, with the largest improvements at the 10 cm level where the non-curriculum policy struggles most.x

| Task | $\pi_{\mathrm{RL}}^1$ | | $\pi_{\mathrm{RL}}^1$ + Curriculum | | $\pi_{\mathrm{base}}^{50}$ | |
|---|---|---|---|---|---|---|
| | 5 cm | 10 cm | 5 cm | 10 cm | 5 cm | 10 cm |
| Alph. Soup + Tom. Sauce | 52.7±5.8 | 14.0±2.0 | **68.7**±5.0 | **28.7**±8.1 | 94.7±5.8 | 68.7±3.1 |
| Both Moka Pots → Stove | 49.3±20.1 | 22.7±8.3 | **70.0**±7.2 | **75.3**±3.1 | 84.0±7.2 | 76.0±7.2 |
| **Average** | 51.0±12.8 | 18.3±4.0 | **69.3**±1.2 | **52.0**±5.3 | 89.3±6.0 | 72.3±2.5 |

**Table XIII:** Position perturbation success rate (%) before and after curriculum training. Curriculum closes much of the gap with $\pi_{\mathrm{base}}^{50}$ without additional demonstrations. Bold marks improvement over non-curriculum $\pi_{\mathrm{RL}}^1$.

These results confirm that progressively expanding the reset distribution during online RL can close much of the position-generalization gap without collecting additional demonstrations. Each curriculum stage builds on the behavioral repertoire acquired at the previous level, allowing the policy to incrementally extend its reach to larger displacements. In Section H, we analyze what drives the position-generalization gap between $\pi_{\mathrm{RL}}^1$ and $\pi_{\mathrm{base}}^{50}$ and why the curriculum is effective.

## APPENDIX H
## POSITION GENERALIZATION ANALYSIS

*a) Position perturbations expose a gap between ID success and state-shift robustness.:* We isolate this effect using position perturbations at different magnitudes. We compare $\pi_{\mathrm{RL}}^1$ to $\pi_{\mathrm{base}}^1$ and to $\pi_{\mathrm{base}}^{50}$, a behavior-cloned policy trained on all 50 human-teleoperated demonstrations for the task. Although $\pi_{\mathrm{RL}}^1$ and $\pi_{\mathrm{base}}^{50}$ achieve similar in-distribution success, Fig. 12b shows that $\pi_{\mathrm{base}}^{50}$ generalizes much better as perturbations grow. *The gap is not explained by the residual parameterization alone.* One possibility is that $\pi_{\mathrm{RL}}^1$ generalizes poorly because the residual actor is less expressive than the

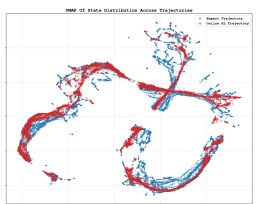

**(a)** State-space coverage.

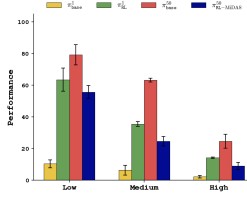

**(b)** Position generalization.

**Figure 12:** **(a)** $\pi_1^{rl}$ covers a narrow region of state space around its warm-start demo; $\pi_{\text{base}}^{50}$ visits a much broader distribution. **(b)** Success degrades with displacement magnitude; distilling $\pi_1^{rl}$ rollouts into the base policy does not close the gap with $\pi_{\text{base}}^{50}$.

**Table XIV:** Success rate (%) of flow-matching BC baselines on LIBERO-Long. Policies are evaluated at the final checkpoint on 50 canonical initial states.

| Task | Image | State |
|---|---|---|
| *LIBERO-Long* | | |
| Alph. Soup + Cr. Cheese $\rightarrow$ Basket | 0.0 | 0.0 |
| Alph. Soup + Tom. Sauce $\rightarrow$ Basket | 0.0 | 0.0 |
| Black Bowl $\rightarrow$ Bottom Drawer | 28.0 | 0.0 |
| Book $\rightarrow$ Caddy | 0.0 | 0.0 |
| Both Moka Pots $\rightarrow$ Stove | 0.0 | 0.0 |
| Cr. Cheese + Butter $\rightarrow$ Basket | 0.0 | 0.0 |
| Moka Pot $\rightarrow$ Stove | 30.0 | 0.0 |
| White Mug + Choc. Pudding | 0.0 | 0.0 |
| White Mug + Plates | 0.0 | 0.0 |
| Yellow-White Mug $\rightarrow$ Microwave | 50.0 | 0.0 |
| **Average** | **10.8** | **0.0** |

pretrained policy architecture. To test this, we distill 50 successful rollouts from $\pi_{\text{RL}}^1$ back into the base policy, yielding $\pi_{\text{RL}-\text{MIDAS}}^{50}$. This still does not recover the robustness of $\pi_{\text{base}}^{50}$; in fact, $\pi_{\text{RL}-\text{MIDAS}}^{50}$ performs worse than $\pi_{\text{RL}}^1$ under position perturbations. This suggests that the gap is driven not by policy parameterization, but by differences in state coverage and motion priors between human expert demonstrations and autonomous data. We visualize this coverage in Fig. 12a, where the 50 human demonstrations span a broader region of state space than successful autonomous trajectories, further supporting this explanation.

## APPENDIX I
## BASELINES

### A. Flow matching policy BC baseline

We include two flow-matching behavior-cloning baselines on LIBERO to separate the effect of the pretrained VLA from the effect of using an expressive action decoder. Both baselines use the same flow-matching action generator and differ only in how observations are represented: one policy conditions on image observations and robot proprioception, while the other conditions on a privileged augmented state vector.

*a) Flow-matching action generator.:* The action generator is a Multi-Modal Diffusion Transformer (MMDiT), following the joint-attention design used in Stable Diffusion 3-style

generative models and adapted here from image generation to action-chunk prediction. We use this architecture as a design template: the flow policy itself is trained on LIBERO demonstrations, while the image-conditioned variant uses ImageNet-pretrained ResNet encoders for visual observations. The policy predicts a chunk of $H = 16$ future actions, where each action is a 7-dimensional end-effector command consisting of $\Delta$position, $\Delta$orientation, and gripper control.

We train the policy with rectified-flow matching. Given an expert action chunk $a$ and Gaussian noise $z \sim \mathcal{N}(0, I)$, we sample a flow time $t \sim \mathcal{U}[0, 1]$ and construct the linear interpolation

$$x_t = tz + (1 - t)a.$$

The network is trained to regress the conditional velocity field along this path using an action-padding-masked MSE loss. At inference time, actions are sampled by integrating the learned velocity field from Gaussian noise using 10 Euler steps, and we use EMA weights for evaluation.

Observation tokens and noised action tokens are linearly embedded to a width of 768 and augmented with frozen 1D sinusoidal positional embeddings. The MMDiT backbone then applies joint attention over observation and action tokens, with the action tokens allowed to attend to all observation tokens. The backbone has 12 transformer blocks, 12 attention heads, head dimension 64, and an FFN expansion factor of 4. Each block uses timestep-conditioned adaLN-Zero modulation, joint multi-head self-attention with RMS-normalized queries and keys, and a feed-forward layer. A final modulated projection maps the action tokens back to $\mathbb{R}^7$.

**Table XV:** Shared hyperparameters for the flow-matching BC baselines.

| Hyperparameter | Value |
|---|---|
| Backbone | MMDiT |
| Transformer depth | 12 |
| Hidden size | 768 |
| Attention heads | 12 |
| Head dim. | 64 |
| FFN expansion | 4 |
| Obs. history | 2 |
| Action horizon | 16 |
| Action dim. | 7 |
| Flow path | rectified flow |
| Flow-time sampling | $\mathcal{U}[0, 1]$ |
| Inference steps | 10 Euler |
| Optimizer | AdamW |
| Learning rate | $10^{-4}$ |
| Weight decay | 0.01 |
| EMA decay | 0.9999 |
| Normalization | z-score |

*b) Image-conditioned flow baseline.:* The image-conditioned baseline uses two camera streams: a third-person `agentview` camera and a wrist-mounted `eye-in-hand` camera. Each camera is processed by an independent ImageNet-pretrained ResNet-18 encoder. We replace BatchNorm layers with GroupNorm and truncate the ResNet at `layer4`, producing a $512 \times 3 \times 3$ feature map per frame. Input images are resized to $100 \times 100$, ImageNet-normalized, and cropped to $84 \times 84$, using random crops during training and center crops during evaluation.

Each camera feature map is converted into a single observation token using a learned spatial-embedding head, followed by a linear bottleneck to the MMDiT width of 768. The policy also receives robot proprioception, consisting of end-effector position, end-effector axis-angle orientation, and gripper joint positions. This 8-dimensional proprioceptive vector is linearly embedded as one additional observation token. Thus, for each observation timestep the image policy receives three tokens: one token for each camera and one proprioceptive token. With a two-step observation history, the policy receives six observation tokens total.

*c) State-conditioned flow baseline.:* The state-conditioned baseline only conditions on a privileged augmented state vector. This vector is used identically when reconstructing observations from dataset MuJoCo states and during online evaluation. The state vector is constructed from four blocks: robot proprioception, object/fixture/site poses, BDDL goal-predicate indicators, and goal-target geometry.

The proprioceptive block contains end-effector position, end-effector axis-angle orientation, and gripper joint positions, giving 8 dimensions. The entity-pose block contains the position and quaternion of every object, fixture, and site, sorted by name. The goal-predicate block contains one binary indicator per BDDL goal sub-predicate. The goal-target geometry block contains the world position of each goal sub-predicate's target entity. For a task with $N_{\text{ent}}$ entities and $N_{\text{goal}}$ goal sub-predicates, the resulting state dimension is

$$D = 8 + 7N_{\text{ent}} + 4N_{\text{goal}}.$$

The state-conditioned policy receives a two-step history of this full augmented state vector and embeds it into observation tokens for the same MMDiT action generator used by the image baseline.

**Table XVI:** Privileged-state dimensions for the LIBERO flow baseline.

| Task | $N_{\text{ent}}$ | $N_{\text{goal}}$ | $D$ |
|------|------|------|------|
| 0 | 17 | 2 | 135 |
| 1 | 17 | 2 | 135 |
| 2 | 7 | 2 | 65 |
| 3 | 13 | 2 | 107 |
| 4 | 10 | 2 | 86 |
| 5 | 11 | 1 | 89 |
| 6 | 10 | 2 | 86 |
| 7 | 11 | 2 | 93 |
| 8 | 7 | 3 | 69 |
| 9 | 9 | 2 | 79 |

### B. Diffusion-Steering RL (DSRL)

DSRL [29] trains a SAC agent that operates in the noise space of the frozen $\pi_{0.5}$ flow-matching policy. At each timestep, the SAC actor receives VLM features extracted by the frozen $\pi_{0.5}$ backbone and outputs a perturbation to the initial noise vector; the frozen denoising head then converts the perturbed noise into an executed action chunk. Because the base policy weights remain fixed, all task-relevant adaptation occurs through the learned noise perturbation. We use the official codebase of Wagenmaker et al. [29] for this baseline.

The SAC actor and critic are three-layer MLPs with hidden dimension 128. Actions are predicted as chunks of length $C = 10$, and online updates begin after 500 initial environment steps. All other training infrastructure (evaluation protocol, reward definition, environment wrappers) matches the MɪDAS Stage II setup (Section E).

**Table XVII:** DSRL hyperparameters.

| Hyperparameter | Value |
|------|------|
| Batch size | 64 |
| Discount factor ($\gamma$) | 0.999 |
| Actor learning rate | $10^{-4}$ |
| Critic learning rate | $3 \times 10^{-4}$ |
| Temperature learning rate | $3 \times 10^{-4}$ |
| Hidden dimensions | $(128, 128, 128)$ |
| Update-to-data ratio | 20 |
| Action chunk length ($C$) | 10 |
| Action noise magnitude | 1.0 |
| Online update start | 500 steps |
| Optimizer | Adam |

### C. Filtered BC

Filtered BC iteratively collects autonomous rollouts from $\pi_{\text{base}}^K$, retains only successful trajectories, and retrains the base policy on this filtered data. The model architecture and trainable parameters are identical to Stage I behavior cloning (Section E-B).

**Procedure.** Training proceeds in rounds. Each round consists of three phases: (1) collect $N$ rollouts using the current policy, (2) filter for trajectories that reached the task goal, and (3) retrain the policy on the filtered set for $M$ gradient steps. The updated checkpoint from each round initializes the next. We use cumulative filtering, where successful trajectories accumulate across rounds so the training set grows monotonically. The $K$ expert demonstrations are included alongside the self-collected successes in every round. We run 20 rounds with 20 rollouts per round and 500 gradient steps per round, for a total of 400 collection rollouts and 10,000 gradient steps.

**Trainable parameters.** The same parameter groups are updated as in Stage I: LoRA adapters (rank 32, $\alpha = 32$) on the PaliGemma 2B vision-language backbone, full fine-tuning of the Gemma 300M action expert, and full fine-tuning of the SigLIP vision encoder. All other backbone weights remain frozen.

**Optimization.** Each round trains with the flow-matching loss using the optimizer and learning-rate schedule from the $\pi_{0.5}$ training config (AdamW, $\beta_1 = 0.9$, $\beta_2 = 0.95$, peak learning rate $2.5 \times 10^{-5}$, cosine decay to $2.5 \times 10^{-6}$). The batch size is 16.

### APPENDIX J
### REAL WORLD IMPLEMENTATION DETAILS

For our real world experiments, we use a bimanual YAM platform and evaluate MɪDAS on two tasks: **(T1)** placing a green block in the right container and a blue block in the left container, and **(T2)** placing a knife and a donut on a plate. We

**Table XVIII:** Filtered BC hyperparameters.

| Hyperparameter | Value |
|---|---|
| Number of rounds | 20 |
| Rollouts per round | 20 |
| Training steps per round | 500 |
| Total training steps | 10,000 |
| Batch size | 16 |
| Cumulative data | Yes |
| Include expert demos | Yes |
| Optimizer | AdamW |
| Peak learning rate | $2.5 \times 10^{-5}$ |
| LR schedule | Cosine decay |
| Minimum learning rate | $2.5 \times 10^{-6}$ |
| LoRA rank | 32 |
| LoRA $\alpha$ | 32 |
| LoRA targets | Attention + FFN |
| Action expert | Fully finetuned |
| Vision encoder (SigLIP) | Fully finetuned |
| Action horizon ($C$) | 10 |
| Loss | Flow matching |

**Table XIX:** Real-world behavior cloning hyperparameters for Stage I finetuning on the YAM platform.

| Hyperparameter | Value |
|---|---|
| Number of demonstrations ($K$) | 1 |
| Training steps | 20,000 |
| Peak learning rate | $2.5 \times 10^{-5}$ |
| LR schedule | Cosine decay |
| LR warmup steps | 1,000 |
| Minimum learning rate | $2.5 \times 10^{-6}$ |
| LR decay steps | 10,000 |
| EMA decay | None |
| PaliGemma 2B backbone | LoRA |
| Action expert (Gemma 300M) | LoRA |
| Action horizon ($C$) | 60 |
| Action type | Delta (joints), absolute (grippers) |
| Cameras | Top, left wrist, right wrist |
| Checkpoint save interval | 5,000 |

fine-tune the $\pi_{0.5}$ base model for 20,000 gradient steps using LoRA adapters for both the vision-language backbone and the action expert. The key differences from the simulation setup (Section E-B) are described below.

**Observation space.** The robot receives RGB images from three cameras: a top-down view and two wrist-mounted cameras (left and right). The state observation consists of the joint positions and gripper states of both arms.

**Action space.** Actions are predicted as chunks of horizon $C=60$, longer than the simulation horizon of 10 to accommodate the lower control frequency and extended task durations on hardware. Joint position targets use delta actions while gripper commands are absolute, matching the bimanual action space structure (6 joints + 1 gripper per arm).

**Trainable parameters.** Unlike the simulation configuration where the Gemma 300M action expert is fully finetuned (Section E-B), the real-world setup applies LoRA adapters to both the PaliGemma 2B vision-language backbone and the Gemma 300M action expert. This reduces the number of trainable parameters and limits overfitting to the single available demonstration per task.

**Optimization.** The learning rate follows a cosine decay schedule with 1,000 warmup steps, a peak learning rate of $2.5 \times 10^{-5}$, and a minimum learning rate of $2.5 \times 10^{-6}$ reached after 10,000 decay steps. EMA averaging is disabled. Table XIX lists all hyperparameters.

**Stage II: Online residual RL.** For Stage II, the residual actor-critic is trained on top of the frozen BC-finetuned policy using PARL (Section E-A). The architecture follows the simulation setup with adjustments for the bimanual action space ($d_a=14$, comprising 6 joints and 1 gripper per arm) and the longer action horizon. The actor and critic are 4-layer MLPs with 1024 hidden units per layer, wider than the 3-layer 512-unit networks used in simulation. The residual policy is queried every 30 steps within each 60-step base action chunk.

Several PARL hyperparameters are scaled relative to the simulation configuration (Table X). The PA-RL procedure samples 32 candidates and retains 10 elites, and the critic and actor receive 40 and 20 updates per gradient step, respectively. A per-joint trust-region constraint of 0.2 rad limits the distance between refined actions and the base policy proposal for non-gripper dimensions; grippers are uncapped. Table XX lists all Stage II hyperparameters.

**Table XX:** Real-world PARL hyperparameters for Stage II online RL on the YAM platform. Entries that differ from the simulation defaults (Table X) are marked with [†].

| Hyperparameter | Value |
|---|---|
| Batch size[†] | 128 |
| Discount factor ($\gamma$)[†] | 0.9999 |
| Target network update rate ($\tau$) | 0.05 |
| Actor learning rate | $10^{-4}$ |
| Critic learning rate | $3 \times 10^{-4}$ |
| Hidden dimensions[†] | $(1024, 1024, 1024, 1024)$ |
| VLM embedding bottleneck dim | 200 |
| Number of Q-networks | 10 |
| Critic ensemble reduction | mean |
| PA-RL candidate samples ($N$)[†] | 32 |
| PA-RL elite samples ($M$)[†] | 10 |
| PA-RL gradient steps | 30 |
| PA-RL step size | $10^{-3}$ |
| PA-RL trust-region radius[†] | 0.2 rad (joints), uncapped (grippers) |
| Critic updates per grad step[†] | 40 |
| Actor updates per grad step[†] | 20 |
| BC warmup steps | 10,000 |
| BC warmup critic/actor updates | 8 |
| Success buffer ratio ($\rho_{\text{succ}}$)[†] | 0.20 |
| Success buffer minimum size[†] | 100 |
| Gradient norm clip | 1.0 |
| Optimizer | Adam |
| Action chunk length ($C$) | 60 |
| Residual query frequency | 30 |
| Action dimension ($d_a$)[†] | 14 |

## APPENDIX K
## FUTURE WORK

Future work can study how far MDA extends beyond the settings tested here. One direction is to test MDA with world-action models, which may provide a different prior from VLA policies through video representations which are predictive over progress, failure, and recoverability. Another boundary is long-horizon tasks, where our RoboCasa results suggest that minimal-demo BC still suffers from distribution shift and compounding error: failures can occur far from terminal reward, and early choices can determine whether later subgoals remain reachable. Finally, the position-perturbation and curriculum results suggest that MDA can guide data collection by distinguishing shifts that require new demonstrations from shifts that can be covered through expanding the reset distribution and online RL.