# OpenReview forum: "Adaptation of Generalist Robot Policies with Minimal Data"
_roboticsfoundation.org/RSS/2026/Workshop/RL4VLA — RL4VLA_

### Official Review · Reviewer_JXiM · 2026-06-22

**Rating:** 6
**Confidence:** 4

**Review:**

The paper introduces MiDAS, an algorithm for refining Generalist Robot Policies with very few demonstrations using residual RL. Specifically, they first fine-tune the base policy using K=1 demonstration and then refine it using a policy conditioned on the base policies, which predicts a full action chunk. They overcome the challenges posed by the few demonstrations problem by initializing the replay buffer using the demonstrations and online samples from just the base policy, initializing the actor to follow the base policy, sampling from the regular online buffer and a success-only online buffer and apply PA-RL to enable the actor to produce actions outside of the narrow mode given by the few demonstrations.

**Strengths:**
- The paper introduces effective measures to stabilize the otherwise brittle residual RL formulation.
- The paper is clearly organized, provides an adequate amount of background, and contains experimental results to back up its claims.

**Weaknesses:**
- Presentation: Captions explaining the figures in more detail are missing for multiple Figures i.e. it's missing what the plot shows (not just how it's results can be interpreted).
- The paper would benefit from experimental results using non-PA-RL residual RL baselines to provide a more thorough basis to compare to.
- Finally, one limitation is that the actor has to predict an entire action chunk of corrections in one call, which could be a limiting factor when scaling to more complex tasks given the size of the actor

---

### Official Review · Reviewer_FTDu · 2026-06-29
**Promising approach with various evaluation, clarity and presentation can be improved**

**Rating:** 6
**Confidence:** 4

**Review:**

**Summary**
The paper proposes MiDAS, a offline to online framework to anchor task-relevant behaviour to an VLA in order to train a residual RL policy with minimal supervision.

**Strengths**
Good motivation, Real World experiments and extensive evaluations.

**Weaknesses**
 Paper is hard to follow. (See comments)

**Comments**
Data usage between methods (table 1) is unclear. e.g. how many successful trajectories are used to train Filt. BC? Adding a RL baseline would strengthen the authors claims. Some Images are small and hard to read. Figure 8 is not discussed in the main paper.

---

### Decision · Program_Chairs · 2026-07-03

**Decision:**

Accept

**Comment:**

This paper presents MIDAS, an offline-to-online reinforcement learning framework that adapts pretrained vision-language-action (VLA) policies from as little as a single demonstration by combining behavior cloning with value-based residual reinforcement learning. The reviewers were generally positive about the paper, with the main concerns being the clarity of the figures and tables, and the lack of discussion on some aspects of the method. We agree with the reviewers' assessment and do not believe that these concerns are sufficient to outweigh the strengths of the paper. We encourage the authors to improve the quality of the figures, tables, and their captions in the camera-ready version, and to address the discussion points and limitations raised by the reviewers.